# Why conspiracy theorists are not always paranoid: Conspiracy theories and paranoia form separate factors with distinct psychological predictors

**Azzam Alsuhibani**[1], **Mark Shevlin**[2], **Daniel Freeman**[3], **Bryony Sheaves**[3], **Richard P. Bentall**[4]*

**1** Department of Psychology, King Saud University, Riyadh, Saudi Arabia, **2** Department of Psychology, University of Ulster, Ulster, United Kingdom, **3** Department of Psychiatry, University of Oxford, Oxford, United Kingdom, **4** Clinical Psychology Unit, Department of Psychology, University of Sheffield, Sheffield, United Kingdom

* r.bentall@sheffield.ac.uk

**Data Availability Statement:** All relevant data are within the paper and its Supporting Information files.

## Abstract

Paranoia and belief in conspiracy theories both involve suspiciousness about the intentions of others but have rarely been studied together. In three studies, one with a mainly student sample (N = 496) and two with more representative UK population samples (N = 1,519, N = 638) we compared single and two-factor models of paranoia and conspiracy theories as well as associations between both belief systems and other psychological constructs. A model with two correlated factors was the best fit in all studies. Both belief systems were associated with poor locus of control (belief in powerful others and chance) and loneliness. Paranoid beliefs were specifically associated with negative self-esteem and, in two studies, insecure attachment; conspiracy theories were associated with positive self-esteem in the two larger studies and narcissistic personality traits in the final study. Conspiracist thinking but not paranoia was associated with poor performance on the Cognitive Reflection Task (poor analytical thinking). The findings suggest that paranoia and belief in conspiracy theories are distinct but correlated belief systems with both common and specific psychological components.

## Introduction

In a seminal essay written during the McCarthy period, the American historian Richard Hofstadter [1] described the persistence of a 'paranoid style' in American politics. Hofstadter gave numerous examples of this style from the history of his own country (for example, panics about the activities of the Illuminati in the 18th century and about the Freemasons in the 19th) but noted that it is not monopolised by any particular nation nor any particular political ideology. This style is, arguably, still discernible in modern political discourse but communism no longer figures prominently amongst the threats to our way of life, which Appelrouth [2] now

**Funding:** This project was supported by a Ph.D. studentship for Dr. Alsuhibani awarded by The Saudi Arabian Cultural Bureau (SACB) and the award number was 17172730.

**Competing interests:** The authors have declared that no competing interests exist.

suggests "come in a variety of forms, some old, some new: 'Islamofascists', homosexuals, liberals, illegal aliens, feminists, the mainstream media".

Although Hofstadter said that he had, "neither the competence nor the desire to classify any figures of the past or present as certifiable lunatics" he argued, nonetheless, that it was reasonable to borrow the term 'paranoid' from the clinical literature. The style, he argued, had to do with "the way in which ideas are believed and advocated rather than with the truth or falsity of their content" so that "the feeling of persecution is central, and it is. . .. systematized in grandiose feelings of conspiracy". His essay and subsequent commentaries have therefore assumed at least a close parallel or even equivalence between the forms of paranoia observed in psychiatric patients and the conspiracy theories that characterise extremist political thinking, implying that belief in conspiracies might be a subclinical variant of the kind of paranoia observed in the psychiatric clinic.

The only difference that Hofstadter acknowledged was that, "the clinical paranoid sees the hostile and conspiratorial world in which he feels himself to be living as directed specifically against him; whereas the spokesman of the paranoid style finds it directed against a nation, a culture, or a way of life". As Imhoff and Lamberty [3] have recently noted, this difference – that paranoia concerns threats to the individual whereas conspiracies are explanations that attribute important events to secret plots by powerful groups – implies that the former may be linked to interpersonal vulnerability and the latter linked to distrust in political institutions. However, because theoretical accounts of paranoia [4, 5] and conspiracy theories [6] have developed separately with very little cross-referral between the two literatures, there have been few attempts to study the relationship between the two phenomena.

## Paranoid beliefs

Paranoid delusions, characterised by the belief that others may be intending harm to the individual, are a common symptom of severe mental illness associated with significant distress in patients with psychosis (schizophrenia). However, sub-clinical paranoia is widely experienced [7] with studies indicating that up to 20 per cent of healthy individuals show significant paranoid ideation [8]. Psychometric evidence confirming that clinical paranoia lies at the extreme end of a continuum with non-clinical paranoid beliefs [9, 10] might be thought to provide support for Hofstadter's thesis.

There is compelling evidence that paranoia, both in population and clinical samples, is associated with early life adversity – especially disrupted attachment bonds – and also adverse socioeconomic circumstances [11]. It has been argued that pathological beliefs in general require two types of processes: emotional factors that determine content and cognitive impairments that prevent individuals from recognising that their beliefs are ill-founded [12]. Consistent with this framework, well established models of the psychological mechanisms responsible for paranoia, although differing in detail, emphasize both types of processes [4, 5]. On the emotional side, an external locus of control [13, 14], negative beliefs about the self [15] and insecure attachment styles [16, 17] have all been associated with paranoia in both clinical and non-clinical populations. Cognitive impairments that have also been implicated in both paranoid patients and in deluded patients in general include theory of mind impairments and a highly replicated tendency to 'jump-to-conclusions' when making judgments about sequentially presented data [18, 19].

## Conspiracy mentality

Like paranoia, conspiracist thinking is common in the general population. In a recent analysis of the US National Comorbidity Study-Replication (N = 5645), we found that 26.7 per cent

agreed with the item, "I am convinced there is a conspiracy behind many things in the world"; endorsement of this item was associated with lower educational achievement and earnings, being outside the labour force, belonging to a minority ethnic group, poor well-being and low social capital [20]. Because conspiracy theories are often shared, they may have important political consequences. For example, their endorsement is associated with the rejection of socially important scientific theories, such as the theory that global warming is the consequence of human use of fossil fuels [21].

Although many specific conspiracy theories are in wide circulation (e.g. the Apollo Moon landings were faked; the British secret service murdered Princess Dianna) there is evidence of a general disposition to believe in conspiracy theories, sometimes referred to as conspiracy mentality [22, 23]. Hence, people who believe in one conspiracy theory are likely to believe in other, even contradictory, conspiracy theories, for example that it is plausible that Princess Diana faked her own death and also that she was killed by the British government [24]. A recent psychological model by Douglas and colleagues suggests that this disposition reflects a combination of epistemic, existential and social motivations [25]. For example, with respect to epistemic needs, it has been shown that individuals who have a high need for closure seize on conspiratorial explanations for uncertain events when such explanations are readily available [26]. Existential challenges such as anxiety-provoking situations [27] or the experience that life is uncontrollable also appear to lead to greater willingness to believe in conspiracies [28, 29]. With respect to social motives, some studies have found that conspiracy theories are associated with individual and collective narcissism [30, 31] and, consistent with this last observation, it has been found that people high in conspiracy mentality have a need to feel unique, and are more likely to endorse conspiracy theories if they think that they were endorsed only by a minority [3]. However, other individual differences that have been implicated in conspiracy theories appear to mirror those thought to be important in paranoia. For example, it has been reported that conspiracy theories are associated with poor analytic thinking [32, 29], and that, like paranoia, they are associated with insecure attachment [20, 33].

## Empirical studies of the association between paranoia and conspiracy theories

Only a modest number of studies have empirically investigated the relationship between paranoia and conspiracy theories. In a small sample (N = 50) of Polish students [34] it was found that endorsement of conspiracy theories about Jews, Arabs, Germans, and Russians correlated with scores on Fenigstein and Vanable's [35] paranoia scale. A study with 120 students, found that conspiracy mentality, paranoia and schizotypy (but not paranormal beliefs) were intercorrelated [36]. Bruder and colleagues [22], in a validation study for their Conspiracy Mentality Questionnaire (see below), found that it correlated strongly with the Fenigstein and Vanable scale ($r$ = .40, N = 120) and Brotherton and colleagues [37] reported a strong correlation between paranoia and scores on their Generic Conspiracist Beliefs Scale ($r$ = .52, N = 150). Wilson and Rose [38] reported modest associations between paranoia and belief in conspiracy theories in a number of student samples ($r$ varying between .27 and .30). Cichocka and collegaues [39], using data from three online studies (Ns varying from 202 to 505) found that conspiracy theories were associated with narcissistic personality traits; although a positive correlation was found between conspiracy theories and paranoia, this effect was explained by paranoia mediating between low self-esteem and conspiracy theories. Finally, in our analysis of the NCS-R data, we also found a positive association between paranoid beliefs and scores on the one-item conspiracy beliefs measure, but associations between insecure attachment and conspiracy theories survived even when paranoia was controlled for [20].

In a meta-analysis of 11 datasets from seven of these studies (plus one study not considered here because a nonspecific measure of schizotypy rather than paranoia was employed), Imhoff and Lamberty [3] found considerable heterogeneity in the data, but the overall effect size was equivalent to a correlation of r = .36 between paranoia and belief in conspiracy theories. In a complex analysis of data from an online survey of 209 German participants using three different measures of paranoia and three measures of belief in conspiracy theories, inter-correlations within the constructs (e.g., different measures of paranoia) were stronger than those between them and a two correlated factor model was found to best fit the data. The latent paranoia variable was found to correlate more with personal variables (e.g., high neuroticism, the tendency to feel egocentric threat) whereas the conspiracy belief factor correlated with politically relevant measures (e.g., low trust in government). These findings were broadly replicated in larger sample (N = 390) of US citizens.

## Purpose of the present study

Here we report three studies in which we build on Imhoff and Lamberty's [3] findings by testing the relationship between paranoia and conspiracy theories, first in a large student sample and then in two large samples representative of the UK population. The studies had two broad aims. First, we aimed to determine whether paranoia and belief in conspiracy theories are separate phenomena. In all three studies we therefore used confirmatory factor analysis to compare models in which paranoia and conspiracy theories are treated as a single construct and those in which they are treated as separate but correlated constructs, predicting that the latter would be a better fit. To test the robustness of the findings, we used two different measures of paranoia and two different measures of belief in conspiracy theories across the three studies, and in our final study we also took steps to eliminate possible method effects attributable to differences in item design.

Second, if paranoia and conspiracy theories are independent but correlated phenomena, it seems likely that some of the psychological processes which explain these propensities are shared whereas others are unique to each. We therefore tested associations between both paranoia and conspiracy theories with psychological constructs that have previously been shown to be important in either one or both of them. Based on existing research on the relationship between self-esteem and paranoia [15], and Imhoff and Lamberty's [3] argument that paranoia is uniquely associated with interpersonal vulnerability, we expected that paranoia would be more closely associated with negative beliefs about the self than conspiracy theories. Past research suggests that both belief systems will both be associated with an external locus of control [14, 28] and insecure attachment [16, 33]. Given that conspiracy theories have been associated with narcissism [31, 39], and given the previously reported strong association between paranoia and negative beliefs about the self [15], we expected narcissism to be specifically associated with conspiracy theories (Study 3).

In two studies (2 and 3) we considered the relationships between both belief systems and analytical reasoning. Although dual process theories of cognition differ in detail [40] it is now widely accepted that human reasoning processes fall into two main types: type 1 (fast, intuitive and associative) versus type 2 (slow, analytic, rational and propositional) [41], and it seems plausible that thinking about emotionally-laden topics such as conspiracies and paranoia will be cognitively 'miserly', that is to say dominated by type-1 thinking and with limited engagement of more effortful type-2 thinking. Freeman and colleagues [42] reported that high paranoia scores in healthy individuals were associated with self-reported experiential (intuitive, from the gut) reasoning. Swami at al. [32] reported similar findings for belief in conspiracy theories and that priming analytical thinking reduced belief in conspiracies. However, the

associations were modest and Freeman et al. [43] were unable to replicate their finding for paranoia in a clinical sample. A limitation of these studies is that they used a self-report measure of the two types of reasoning whereas an objective measure would be preferable. In our second and third studies we therefore included expanded versions of Frederick's [44] Cognitive Reflection Test (CRT), which measures 'miserly' cognition in which type 1 thinking is not regulated by more effortful type-2 analytical thinking [45]. A modest association between poor performance on the CRT and subclinical paranoia has been reported [46], and a similar association has been reported for conspiracy theories [29]. Hence, we expected that poor CRT performance would be associated with both belief systems in this study.

Finally, in Studies 1 and 2 we examined the relationships between paranoia and belief in conspiracy theories and social relationships, although our predictions were more tentative. On the one hand, because (as Hofstadter pointed out) paranoid beliefs tend to be idiosyncratic whereas conspiracy theories tend to be shared, it might be predicted that paranoid people would be more socially isolated, and hence lonelier, than people who believe in conspiracy theories and, indeed, in a previous study with a student sample we found that paranoia was strongly associated with loneliness [47]. On the other hand, in previous studies, loneliness has been associated with poor interpersonal trust [48] and, in our own previous work, we found that conspiracy theories were associated with low social capital [20]. We therefore included a measure of loneliness in Studies 1 and 2.

## Study 1

### Methods

**Participants and procedure.** Participants received invitations to complete an online survey by emails sent to participant panels at Liverpool, Oxford and Ulster universities. Ethical approval was obtained from the University of Liverpool ethics committee (Reference number, 718/2016). The survey began with an online information page, and participants indicated their informed consent by clicking checkboxes indicating that they had understood the instructions and knew of their right not to participate or to withdraw at any time by closing their browser. No minors were involved in the study.

A total of 790 people participated in the survey, 254 men with a mean age of 27.57 years (SD = 15.48) and 536 women with mean age of 26.05 (SD = 11.76). 569 of the participants were students, 180 employed, and 24 not in employment (17 did not provide data). For the purpose of the present study, we excluded all the participants who completed the survey in less than 15 minutes (to ensure adequate attention to the questionnaires) or who completed less than 50% of the items. Because of the cultural sensitivity of many of the items, we also excluded those who were not born in the UK. The final sample was 496 participants (158 men, mean age = 28.97 years SD=13.75, and 338 women, mean age = 26.13 years, SD = 12.19).

**Measures.** The data considered here formed part of a multipurpose survey which included a wide range of questions on political, religious, social and other kinds of belief systems and relevant psychological constructs.

*The Revised Paranoia and Deservedness Scale* (PaDS –R) was designed on the basis of psychometric analyses of the original scale [49] in a large sample of healthy individuals and patients with psychosis [10] and other recent findings indicating that paranoia consists of four elements: interpersonal sensitivity, mistrust, fear of persecution and ideas of reference [9]. The revised scale consists of 8 paranoia items (the P scale), two each from these domains (e.g. respectively, "My friends often tell me to relax and stop worrying about being deceived or harmed", "You should only trust yourself", "I believe that some people want to hurt me deliberately", and "Sometimes I think there are hidden insults in things that other people say or

do"). Items are answered on a 5-point scale ranging from "Strongly agree" to Strongly disagree". In this sample, the eight-item P scale had an alpha coefficient of .87.

*The Conspiracy Mentality Questionnaire* (CMQ) [22] is a five item scale (e.g. "I think that many very important things happen in the world, which the public is never informed about") assessing participants' general tendency to believe in conspiracies. Responses are on 11-point scales indicating how likely it is that respondents think each of the items is true from 0 ("0% – Certainly not") to 10 ("100% – Certain"). The alpha coefficient in this sample was .84.

*The Relationship Questionnaire* (RQ) [50] was used to assess attachment style. Participants read four vignettes describing secure, fearful, preoccupied and dismissing prototypical styles and have to choose the one that describe them best. They are then asked to rate each vignette "according to how well or poorly each description corresponds to your general relationship style" on 7-point scales from "Disagree strongly" to "Agree strongly". Scores on the four scales can be used to compute higher order measures of attachment anxiety (negative model of self) and attachment avoidance (negative model of other).

*The Brief Core Schema Scale* (BCSS) [51], developed originally to assess processes thought to be important in psychotic phenomena, measures four self-schematic constructs: negative beliefs about the self (e.g. "I am unloved"), positive beliefs about the self (e.g. "I am respected"), negative beliefs about others (e.g. "Other people are hostile") and positive beliefs about others (e.g. "Other people are fair"), each with 6 items rated on 5-point scales ("Don't' believe" to "Believe totally"). In the present study, alpha coefficients for the four subscales ranged from .80 to .93.

*The Multidimensional Locus of Control Scale* (MLCS) [52] has three 8-item subscales: internality (e.g. "Whether or not I get to be a leader depends mostly on my ability"), chance (e.g. "When I get what I want, it's usually because I'm lucky") and powerful others (e.g. "My life is chiefly controlled by powerful others"). Each item is rated on a five-point scale from "Strongly disagree" to "Strongly agree". In the present sample, the internality subscale had an alpha of .64, the chance subscale had an alpha of .72 and the powerful others subscale had an alpha of .76.

*The Loneliness Scale* [53] has three items (e.g. "How often do you feel left out?") answered on 3-point scales ("Hardly ever", "Some of the time", "Often"). In the present sample, the alpha coefficient was .83.

**Statistical approach.** First, confirmatory factor analyses (CFA) were conducted to compare two models: a model in which all paranoia and conspiracy mentality items loaded on a single conspiracy/paranoia factor, and a model in which paranoia and conspiracy mentality are separate but correlated latent variables. Second, after establishing that the two-factor was a better fit, we examined associations with subscales of each of the other psychological constructs of interest (attachment style, self-schemas, locus of control and loneliness). For this purpose, we calculated a regression model in which all of the psychological constructs were entered simultaneously as predictors. In this model, conspiracy mentality (CMQ items) and paranoia (PaDS-R P items) were considered as latent factors, which were allowed to covary. This approach does not require us to partial out common and shared components of paranoia and conspiracist thinking but it does allow us to test for the specificity of the associations between constructs and belief systems. To achieve this, equality constraints were initially placed on the regression coefficients predicting the latent variables; these equality constraints were tested using Wald tests. If a Wald test was significant, the regression coefficients between the construct and paranoia and conspiracy theories were considered to be significantly different.

Confirmatory factor and regression models were conducted in Mplus 7.0 [54] with robust maximum likelihood estimation (MLR; [55]). The following recommendations were followed

to assess model fit [56, 57]: a non-significant chi-square ($\chi^2$), Comparative Fit Index (CFI) [58] and Tucker Lewis Index (TLI) [59] values above .95 reflect excellent fit, while values above for these two indices above .90 reflect acceptable fit; Root-Mean-Square Error of Approximation with 90% confidence intervals (RMSEA) [60] with values of .06 or less reflect excellent fit while values less than .08 reflect acceptable fit. The Standardized Root-Mean-Square Residual (SRMR) [61] was also used with values of .06 or less indicating excellent fit and values less than .08 indicating acceptable fit. The Bayesian Information Criterion (BIC) [62] was used to evaluate and compare models, with the smallest value indicating the best fitting model. In relation to the BIC, it has been suggested that a 2-6 point difference offers evidence of model superiority, a 6-10 point difference indicates strong evidence of model superiority, and a difference greater than 10 points indicates very strong evidence of model superiority [63].

## Results

The correlation matrix for the variables included in the study is shown in Table 1. The Pearson correlation between PaDS-R total scores and CMQ total scores was significant, $r$ = .34, p < .001.

**Model fit for paranoia and conspiracy theories.** The model fit indices for the initial confirmatory factor models, which included only the PaDS-R P and CMS items, showed that the two factor model provided acceptable fit ($\chi^2$ (64) = 204.888, p > .05; RMSEA = .068; CFI = .934; TLI = .919, SRMR = .046) and the one factor model did not ($\chi^2$ (65)=844.474, p >.05; RMSEA = .158; CFI = .634; TLI = .561, SRMR = .126). The BIC was also lower for the two factor model (BIC = 21590.384) compared to the one factor model (BIC = 22300.743) and the difference was much greater than 10 points and so indicates very strong evidence of the superiority of the two factor model. The standardised factor loadings for the paranoia and conspiracy mentality latent variables were all high, positive and statistically significant ranging from .520 to .858, and the correlation between the latent variables was .417. The composite reliability [64] for paranoia (CR = .870) and conspiracy mentality (CR = .841) were high.

**Associations between paranoia, conspiracy theories and psychological constructs.** To assist in the interpretation of the findings, bivariate correlations and partial correlations (controlling for the other belief system) between paranoia and conspiracy mentality and each of the psychological constructs are shown in S1 Table. Partial correlations between belief systems and psychological constructs for the three studies.

Table 2 shows the regression coefficients and Wald tests for our regression model. Both anxious attachment (model of self) and avoidant attachment (model of other) were associated with both belief systems but the effect of anxious attachment was much greater for paranoia than conspiracy mentality. Paranoia was associated with low positive beliefs about the self and high negative beliefs about the self but both systems were associated with negative beliefs about others; the effect for negative beliefs about the self was significantly greater for paranoia than conspiracy mentality. As expected, two of the locus of control subscales – chance and powerful others – were associated with both belief systems, although the effect of chance was greater for paranoia. An association with internality was only found for paranoia although this was not significantly greater than the non-association found for conspiracist thinking. Finally, although both belief systems were associated with loneliness, the effect was much stronger for paranoia.

## Study 1 discussion

We found evidence that paranoia and conspiracist thinking, although correlated to approximately the same extent found in previous studies [3], appear to be distinct phenomena; a

**Table 1. Study 1: Zero-order correlations [and 95% confidence intervals] between paranoia, conspiracy theories, and psychological constructs.**

| Predictor Variables | | 1 | 2 | 3 | 4 | 5 | 6 | 7 | 8 | 9 | 10 | 11 |
|---|---|---|---|---|---|---|---|---|---|---|---|---|
| 1. Paranoia | | _____ | | | | | | | | | | |
| 2. Conspiracy Theories | | .367** [.275,.454] | _____ | | | | | | | | | |
| Attachment | 3. Model of Self | -.476** [-.536, -.406] | -.125* [-.216, -.029] | _____ | | | | | | | | |
| | 4. Model of Others | -.212** [-.304, -.116] | -.111 [-.209, -.007] | .092 [.024, .163] | _____ | | | | | | | |
| Self-schemas | 5. Positive Self | -289** [-.372, -.211] | -.015 [-.119, .081] | .306** [.224, .378] | .106* [.015, .192] | _____ | | | | | | |
| | 6. Negative Self | .397** [.315, .474] | .015 [-.077, .119] | -.397** [-.470, -.325] | .089 [-.189, .007] | | _____ | | | | | |
| | 7. Positive Others | -.199** [-.301, -.102] | -.043 [-.152, .063] | .122* [.037, .216] | .219** [.125, .317] | .423** [.342, .496] | -.188** [-.283, -.095] | _____ | | | | |
| | 8. Negative Others | .407** [.320, .495] | .256** [.146, .357] | -.232** [-.328, -.136] | -.003 [-.101, .094] | -.039 [-.133, .055] | .249** [.155, .339] | -.016 [-.141, .100] | _____ | | | |
| Locus of control | 9. Internality | -.120 [-.223, -.021] | -.028 [-.139, .099] | .179** [.081, .272] | .004 [-.090, .100] | .255** [.153, .356] | -.202** [-.286, -.120] | .210** [.092, .316] | -.141** [-.242, -.039] | _____ | | |
| | 10. Chance | .413** [.331, .488] | .222** [.124, .316] | -.247** [-.334, -.154] | -.048 [-.172, .072] | -.102* [-.199, -.002] | .259** [.169, .352] | -.022 [-.134, .084] | .242** [.156, .332] | -.088 [-.202, .037] | _____ | |
| | 11. Powerful Others | .342** [.251, .430] | .223** [.117, .319] | -.228** [-.321, -.128] | -.001 [-.097, .093] | -.104* [-.208, .001] | .230** [.114, .339] | -.118* [-.227, -.007] | .204** [.109, .310] | .002 [-.127, .127] | .528** [.439, .611] | _____ |
| 12. Loneliness | | .537** [.466, .603] | .148** [.061, .245] | -.500 [-.570, -.421] | -.219** [-.300, -.129] | -.342** [-.430, -.251] | .499** [.414, .573] | -.224** [-.309, -.138] | .305** [.207, .397] | -.176** [-.271, -.084] | .259** [.166, .343] | .248** [.157, .340] |

Note

** $p < .01$

* $p < .05$.

confirmatory model with two correlated factors was a far better fit to the data than a single factor model. The two systems showed differential associations with some psychological constructs but not others: attachment anxiety, negative beliefs about the self, loneliness and (marginally) internality were more associated with paranoia, consistent with Imhoff and Lamberty's [3] account of the difference between the two systems and also with psychological models of paranoia which emphasize the role of low self-esteem and interpersonal sensitivity [4, 5]. The strong association between paranoia and loneliness observed in the present study has been observed in a previous study with a student sample [47].

The study had several limitations. First, the primarily student sample was unrepresentative of the general population. Hence, we decided to attempt to replicate our findings with a larger, more diverse and more representative sample. Second, our conspiracy mentality measure was a short assessment of the tendency to believe in conspiracies in general, so that participants did not have to rate the plausibility of specific conspiracy theories; we decided to replace it with a scale that does. Third, the BCSS, which we used to measure self-schemas, was originally

**Table 2. Study 1: Standardised regression coefficients and tests of equality from multivariate regression model predicting paranoia and conspiracy beliefs.**

| Predictor Variables | | Paranoia β (se) | Conspiracy β (se) | Wald | df | p |
|---|---|---|---|---|---|---|
| Attachment | Model of Self | -.422 (.035) * | -.120 (.045) * | 45.508 | 1 | < .001 |
| | Model of Other | -.153 (.043) * | -.125 (.046) * | 0.306 | 1 | .580 |
| Self-esteem | Positive self | -.122 (.051) * | -.037 (.060) | 1.881 | 1 | .170 |
| | Negative self | .209 (.051) * | -.073 (.056) | 30.545 | 1 | < .001 |
| | Positive other | -.093 (.050) | -.059 (.051) | 0.450 | 1 | .502 |
| | Negative other | .339 (.045) * | .283 (.047) * | 1.461 | 1 | .227 |
| Locus of Control | Internality | -.099 (.048) * | -.013 (.051) | 2.746 | 1 | .098 |
| | Chance | .299 (.052) * | .171 (.060) * | 4.664 | 1 | .031 |
| | Powerful Others | .174 (.052) * | .139 (.058) * | 0.430 | 1 | .512 |
| Loneliness | | .501 (.039) * | .152 (.047) * | 53.968 | 1 | < .001 |
| R-squared | | .456 (.036) * | .152 (.031) * | 27.019 | 1 | <. 001 |

Note

* p < .05.

designed for the purposes of research into psychosis and we therefore replaced it with a more conventional measure of self-esteem. Finally, we included an expanded version of Frederick's [44] Cognitive Reflection Test (CRT) as a measure of analytic thinking/cognitive miserliness.

## Study 2

### Methods

**Participants and procedure.** The study was authorised under the same ethics approval for Study 1, and the consent procedure was identical. Participants were recruited to be a close to representative national sample by the survey company Qualtrics, and were stratified by age (minimum age 18 years; approximately equal numbers from age bands 18-24; 25-34; 45-49; 50-64; 65+), sex and household income (approximately equal numbers from quintiles defined on the basis of Office for National Statistics data (https://www.ons.gov.uk/peoplepopulationandcommunity/personalandhouseholdfinances/incomeandwealth/bulletins/householddisposableincomeandinequality/financialyearending2016): £0-599; £600-1,155; £1,156-2,247; £2,248-£3,604; £3,605-£7.061; £7,062 and above). 1,852 UK residents attempted the survey but, after removal of incomplete surveys or surveys completed implausibly quickly (pre-defined following pilot work by the survey company as < 12 minutes) the final sample was 1,519.

Of these, 753 were men with a mean age of 50.66 years (SD = 18.22) and 766 were women with a mean age of 45.04 years (SD = 15.64). 1451 were British nationals and 68 had other nationality.

**Measures.** The data for this study were again drawn from a multipurpose survey. The following measures were identical to those used in Study 1: *Revised Paranoia and Deservedness Scale* (in this study, alpha = .91); *The Relationship Questionnaire*; The *Multidimensional Locus of Control Scale* (alphas for internality = .75; chance = .82; powerful others = .85); and the *Loneliness Scale* (alpha = .88). The following additional measures were used:

*Generic Conspiracist Beliefs Scale* (GCBS) [65] includes items about a range of conspiracy theories (e.g. "Evidence of alien contact is being concealed from the public" and "Groups of scientists manipulate, fabricate, or suppress evidence in order to deceive the public"). The original scale included 15 items but this version included two additional items based on the work of Wood et al. [24] designed to test whether people high on conspiracy mentality would

endorse mutually contradictory conspiracy theories: "Princess Diana faked her death so that she could retreat into isolation" and "Princess Diana had to be killed because the British government could not accept that the mother of a future king was involved with a Muslim Arab". Responses are rated on 5-point scales ("Definitely not true" to "Definitely true"). The alpha coefficient for the scale was .94.

*Self-esteem rating scale short form* (SERS) [66] is a 20-item scale, designed to assess self-esteem without scores being contaminated by mood which, in confirmatory factor analyses with both nonclinical and clinical (severe mental illness) samples, has been shown to yield two negatively correlated subscales, positive self-esteem (10 positive statements about the self, e.g. "I feel good about myself") and negative-self-esteem (10 negative statements about the self, e.g. "I feel that others do things much better than I do"). Participants rate each statement from 1, 'never', to 7, 'always'. In this study, alpha was 0.94 for positive self-esteem and .94 for negative self-esteem.

*Cognitive Reflection Test* (CRT) [44] is a 3-item scale in which requires an answer to mathematical questions, in which the structure of the question implied the wrong answer, so that the correct answer requires effortful reflection and 'cognitive miserliness' leads to wrong answer e.g. "A bat and a ball cost £1.10 in total. The bat costs £1.00 more than the ball. How much does the ball cost?". A second version of the scale, the CRT-2 [67] included 4-item scale with a lower degree of mathematical complexity, e.g. "A farmer had 15 sheep and all but 8 died. How many are left?", and we combined both measures to make a 7-item scale. Participants typed their answers into a textbox. Only 30 seconds from the moment of presentation was allowed for each answer, after which the questionnaire automatically moved to the next item. The alpha coefficient for 1,235 who completed all 7 items was .71.

## Results

Zero-order correlations between the study variables are shown in Table 3. In this study, the correlation between the summed scores on the PaDS-R and our conspiracy measure, the GCBS, was .44, p < .001.

Replicating Wood et al. [24], a positive correlation was found between participants' ratings of the plausibility of the two contradictory conspiracy theories, "Princess Diana faked her own death so that she could retreat into isolation" and "Princess Diana was killed because the British government could not accept that the mother of the future king was involved with a Muslim Arab", r = .39, p < .001.

**Model fit for paranoia and conspiracy theories.** The model fit indices for the CFA models which included the PaDS-R P and GCBS items showed that the two factor model provided acceptable fit ($\chi^2$ (274) = 2169.513, p >.05; RMSEA = .067; CFI = .892; TLI = .882, SRMR = .051) and the one factor model did not ($\chi^2$ (275) = 5816.359, p >.05; RMSEA = .115; CFI = .685; TLI = .657, SRMR = .120). The BIC was also lower for the two-factor model (BIC = 102145.63) compared to the one factor model (BIC = 107084.68) and indicates very strong evidence of the superiority of the two factor model. The standardised factor loadings for the paranoia and conspiracy mentality latent variables were all high, positive and statistically significant ranging from .488 to .842, and the correlation between the latent variables was .459. The composite reliability for paranoia (CR = .907) and conspiracy mentality (CR =.947) were high.

**Associations between paranoia, CTs and psychological constructs.** Bivariate associations and partial correlations between the belief system variables and psychological constructs are shown in S1 Table. Partial correlations between belief systems and psychological constructs for the three studies.

The regression model and Wald tests of whether the predictor variables are differentially associated with the two types of belief systems are shown in Table 4. In general, the patterns of

**Table 3. Study 2: Zero-order correlations [and 95% confidence intervals] between paranoia and conspiracy mentality and psychological constructs.**

| Predictor Variables | | 1 | 2 | 3 | 4 | 5 | 6 | 7 | 8 | 9 | 10 |
|---|---|---|---|---|---|---|---|---|---|---|---|
| 1. Paranoia | | _____ | | | | | | | | | |
| 2. Conspiracy Theories | | .433** [.382, .490] | _____ | | | | | | | | |
| Attachment | 3. Model of Self | -.413** [-.458, -.365] | -.143** [-.197, -.085] | _____ | | | | | | | |
| | 4. Model of Others | -.178** [-.235, -.123] | .034 [-.026, .093] | .183** [.124, .240] | _____ | | | | | | |
| Self-schemas | 5. Positive | -.244** [-.308, -.181] | .055 [-.014, .117] | .325** [.267, .382] | .213** [.153, .271] | _____ | | | | | |
| | 6. Negative | .739** [.711, .763] | .345** [.289, .400] | -.409** [-.456, -.359] | -.084** [-.137, -.029] | -.268** [-.343, -.198] | _____ | | | | |
| Locus of control | 7. Internality | -.035 [-.099, .031] | .064* [-.002, .120] | .183** [.130, .234] | .094** [.041, .145] | .490** [.429, .547] | -.115** [-.183, -.054] | _____ | | | |
| | 8. Chance | .529** [.477, .578] | .426** [.370, .480] | -.240** [-.288, -.190] | -.053 [-.113, .005] | .008 [-.059, .073] | .505** [.449, .559] | .227** [.150, .293] | _____ | | |
| | 9. Powerful Others | .556** [.509, .602] | .418** [.362, .471] | -.278** [-.328, -.221] | -.030 [-.083, .024] | -.037 [-.104, .033] | .552** [.497, .602] | .236** [.171, .296] | .787** [.757, .817] | _____ | |
| 10. Loneliness | | .588** [.550, .625] | .281** [.229, .332] | -.406** [-.452, -.359] | -.132** [-.188, -.077] | -.341** [-.396, -.284] | .620** [.582, .659] | -.142 [-.202, -.089] | .345** [.296, .396] | .355** [.304, .405] | _____ |
| Cognitive Reflection Test | 11. Number correct | -.095** [-.151, -.043] | -.233** [-.286, -.182] | .023 [-.033, .076] | -.041 [-.100, .016] | -.057* [-.111, -.003] | -.091** [-.151, -.034] | -.037 [-.091, .027] | -.111** [-.164, -.058] | -.097** [-.150, -.040] | -.033 [-.092, .032] |

Note

** $p < .01$

* $p < .05$.

**Table 4. Study 2: Standardised regression coefficients and tests of equality from multivariate regression model predicting paranoia and conspiracy beliefs.**

| Predictor Variables | | Paranoia β (se) | Conspiracy β (se) | Wald | df | p |
|---|---|---|---|---|---|---|
| Attachment | Model of Self | -.069 (.022) * | .004 (.030) | 5.778 | 1 | .016 |
| | Model of Other | -.092 (.020) * | .040 (.026) | 20.571 | 1 | < .001 |
| Self-esteem | Positive | -.024 (.025) | .145 (.033) * | 19.247 | 1 | < .001 |
| | Negative | .461 (.031) * | .081 (.039) * | 63.389 | 1 | < .001 |
| Locus of Control | Internality | .022 (.025) | -.080 (.030) * | 9.214 | 1 | .002 |
| | Chance | .111 (.032) * | .202 (.043) * | 4.576 | 1 | .032 |
| | Powerful Others | .116 (.034) * | .175 (.045) * | 1.636 | 1 | .201 |
| Loneliness | | .180 (.026) * | .146 (.034) * | 0.736 | 1 | .391 |
| Cognitive Reflection Test | Correct | -.025 (.019) | -.173 (.024) * | 33.788 | 1 | < .001 |
| R-squared | | .620 (.017) * | .274 (.024) * | 146.534 | 1 | < .001 |

Note: $p < .05$*.

association are similar to those observed in Study 1, but with higher significance. Both insecure attachment styles are associated with paranoia and not conspiracy theories. Paranoia but not conspiracy theories are associated with negative self-esteem whereas the opposite is true of positive self-esteem. For the locus of control subscales, both types of belief systems are associated with belief in powerful others and chance, although the association for chance is greater for conspiracy theories than for paranoia (the reverse was the case in Study 1); only conspiracy theories are associated with low internality scores (in Study 1 paranoia was associated with low internality).

The findings for social capital are less consistent. Loneliness is equally associated with both belief systems whereas there is a modest positive association between conspiracy theories and network diversity (the number of high contact roles) and a modest negative association between conspiracy theories and the social network index (total network size). Finally, conspiracy theories are uniquely associated with miserly analytical thinking as measured by the CRT.

## Study 2 discussion

Replicating Study 1 and previous studies [3], paranoia and belief in conspiracy theories, although correlated, were best modelled as two separate constructs. Also replicating Study 1, paranoia was more closely associated than belief in conspiracies with psychological constructs indicative of interpersonal vulnerability, specifically insecure attachment and negative beliefs about the self. Both chance and powerful others locus of control were associated with both belief systems, again as in Study 1; the inconsistent finding for internality (negatively associated with paranoia in Study 1 but with conspiracy theories in Study 2) should be interpreted in the context of the small magnitude of the effects in both studies. As in Study 1, both paranoia and conspiracy theories were associated with loneliness but the effect for paranoia was much less, perhaps reflecting the fact that Study 2 employed a sample that was much more representative of the general population.

The finding that poor performance on the CRT was specifically related to belief in conspiracy theories is striking. Previous studies have reported that impaired analytic thinking is associated with both belief systems [46, 29] but no previous study has considered both belief systems together.

Although Study 2 represented a methodological advance on Study 1, particularly in terms of the sample, we decided to conduct a third study to address some remaining questions. The first concerned a potential methodological limitation affecting both Study 1 and Study 2. We wondered whether the superiority of a two-factor model over a single factor model of paranoia and belief in conspiracy theories could reflect a method effect related to item format. Items in both the Conspiracy Mentality Scale employed in Study 1 ("I think that many very important things happen in the world, which the public is never informed about") and Generic Conspiracist Beliefs Scale employed in Study ("Groups of scientists manipulate, fabricate, or suppress evidence in order to deceive the public") required participants to indicate their agreement with specific propositions about events in the world where as some of the Items in the Paranoia and Deservedness Scale ("My friends often tell me to relax and stop worrying about being deceived or harmed") less directly assessed beliefs and include affective elements. We therefore created a new paranoia scale, with exactly the same response format as the GCBS, in which each item was entirely propositional.

Second, although both Studies 1 and 2 showed a specific association between insecure attachment and paranoia, the scale we used to measure attachment, the Relationship Questionnaire, was not that used by Green and Douglas [33] in their study showing a relationship

between attachment insecurity on conspiracy theories; in Study 3 we therefore used the ECR employed in their study.

Third, we sought to replicate our finding that impaired performance on the CRT is specifically associated with belief in conspiracy theories. Finally, given previous findings that belief in conspiracy theories is associated with narcissism [31, 39] we included a narcissism measure.

## Study 3

### Methods

**Participants and procedure.** The study was authorised under the same ethics approval for Study 1, and the consent procedure was identical. Participants were recruited by the survey company Qualtrics using the same sampling frame employed in Study 2, with participants stratified by sex, age and household income. A total of 722 UK residents attempted the survey but, after removal of incomplete surveys or surveys completed implausibly quickly (pre-defined following pilot work by the survey company as < 15 minutes) the final sample was 638.

Of these, 296 were men with a mean age of 46.60 years (SD = 15.83) and 342 were women with a mean age of 43.77 years (SD = 16.16).

### Measures

The data for this study were again drawn from a multipurpose survey. The following measures were identical to those used in Study 1: The *Multidimensional Locus of Control Scale* (alphas in this study for internality = .79; chance = .84; powerful others = .86); the *Self-Esteem Rating Scale* (alphas for positive self = .94; negative self = .95). The study also used the 15-item version of the *Generic Conspiracist Beliefs Scale* employed in Study 2 (alpha = .96) (the two additional items from Study 2 designed to assess contradictory conspiracy theories were not included).

The revised *Paranoia Scale* designed especially for this study was based on the PaDS employed in Studies 1 and 2, but items were rewritten so that each contained a specific proposition formatted similarly to the items of the GCBS, and with an identical response format. There were two items for each of the domains identified by Bebbington et al. [9]: interpersonal sensitivity ("There is a risk that I will be criticised or rejected in social situations"); mistrust ("You should only trust yourself"); ideas of reference ("When I am out in public, people sometimes talk about me") and fear of persecution ("Some people want to hurt me deliberately"). Coefficient alpha for the scale was .85.

*Cognitive Reflection Test* (CRT) [44] was expanded to include 7 items from two previous alternative versions of the test [45, 67]. This version used the four-option multiple choice format with choices presented in random order as recommended by Sirota and Juanchich [68]; 45 seconds was allowed for each answer, after which the questionnaire automatically moved to the next item. The alpha coefficient for 597 who completed all 10 items was .70.

*A brief version of the Experiences in Close Relationships Scale* (ECR-12) [69] is a 12-item scale used for the assessment of two attachment styles: attachment anxiety (e.g. "I worry that others won't care about me as much as I care about them") and avoidant attachment (e.g. "I feel comfortable depending on others"). Responses are rated on 7-point scale, from ("strongly disagree" to "strongly agree"). The alpha coefficients for anxious attachment was .87 and for avoidant attachment was .77.

*The Narcissistic Personality Inventory* (NPI-13) [70] has 13-items in which participants are presented with pairs of attributes scale and have to choose the one they most agree with (e.g. "A- I find it easy to manipulate people. B- I don't like it when I find myself manipulating people"). The alpha coefficient for the scale was .77.

## Results

Zero-order correlations between the study variables are shown in Table 5. In this study, the correlation between the summed scores on the paranoia and the conspiracy measures, was .50, p < .001.

**Model fit for paranoia and conspiracy theories.** The model fit indices for the CFA models which included the Revised Paranoia Scale and GCBS items showed that the two factor model provided acceptable fit ($\chi^2$ (229) = 813.70, p < .001; RMSEA = .063; CFI = .913; TLI = .904, SRMR = .055) whereas the one factor model did not ($\chi^2$ (230) = 1656.88, p < .001; RMSEA = .099; CFI = .788; TLI = .767, SRMR = .097). The BIC was also lower for the two-factor model (BIC = 38336.09) compared to the one factor model (BIC = 39508.56), providing strong evidence of the superiority of the former. The standardised factor loadings for the paranoia and conspiracy mentality latent variables were all positive and statistically significant ranging from .302 to .837, and the correlation between the latent variables was .530. The composite reliability for paranoia (CR = .855) and conspiracy beliefs (CR =.956) were high.

**Associations between paranoia, conspiracy theories and psychological constructs.** Bivariate associations and partial correlations between the belief system variables and

**Table 5. Study 3: Zero-order correlations [and 95% confidence intervals] between Paranoia and conspiracy mentality and psychological constructs.**

| Predictor Variables | | 1 | 2 | 3 | 4 | 5 | 6 | 7 | 8 | 9 | 10 |
|---|---|---|---|---|---|---|---|---|---|---|---|
| 1. Paranoia | | _____ | | | | | | | | | |
| 2. Conspiracy Theories | | .503** [.433, .569] | _____ | | | | | | | | |
| Attachment | 3. Avoidant Attachment | .034 [-.041, .122] | .018 [-.063, .101] | _____ | | | | | | | |
| | 4. Attachment Anxiety | -.008 [-.089, .075] | .033 [-.044, .117] | -.158** [-.241, -.062] | _____ | | | | | | |
| Self-schemas | 5. Positive | -.055 [-.149, .039] | .033 [-.055, .120] | -.035 [-.116, .053] | .041 [-.036, .117] | _____ | | | | | |
| | 6. Negative | .454** [.369, .530] | .437** [.360, .512] | .028 [-.084, .110] | -.020 [-.111, .068] | -.206** [-.305, -.108] | _____ | | | | |
| Locus of control | 7. Internality | .117** [.011, .222] | .068 [-.031, .158] | -.024 [-.111, .062] | -.020 [-.101, .062] | .462** [.376, .537] | -.010 [-.120, .096] | _____ | | | |
| | 8. Chance | .410** [.318, .502] | .454** [.368, .536] | -.017 [-.098, .061] | -.039 [-.119, .041] | -.024 [-.130, .071] | .552** [.479, .622] | .299** [.191, .394] | _____ | | |
| | 9. Powerful Others | .433** [.335, .518] | .459** [.368, .540] | -.024 [-.111, .057] | -.024 [-.097, .048] | -.050 [-.153, .051] | .612** [.541, .672] | .303** [.194, .399] | .791** [.746, .832] | _____ | |
| 10. Narcissism | | .212** [.139, .284] | .313** [.242, .381] | -.013 [-.094, .064] | .037 [-.046, .120] | .234** [.160, .304] | .264** [.185, .346] | .073 [-.014, .161] | .111** [.012, .209] | .170** [.073, .262] | _____ |
| Cognitive Reflection Test | 11. Number correct | -.094* [-.175, -.019] | -.273** [-.340, -.191] | -.055 [-.130, .021] | .040 [-.049, .129] | -.106** [-.186, -.028] | -.127** [-.201, -.049] | .035 [-.039, .114] | -.059 [-.136, .018] | -.027 [-.095, .040] | -.168** [-.244, -.088] |

Note:** p < .01

* p < .05.

**Table 6. Study 3: Standardised regression coefficients and tests of equality from multivariate regression model predicting paranoia and conspiracy beliefs.**

| Predictor Variables | | Paranoia β (se) | Conspiracy β (se) | Wald | df | p |
|---|---|---|---|---|---|---|
| **ECR-12 Attachment** | Attachment Avoidance | .030(.052) | .006(.051) | 0.075 | 1 | .785 |
| | Attachment Anxiety | -.024(.048) | .048(.048) | 0.698 | 1 | .404 |
| **Self-esteem** | Positive | -.157(.054) * | .130(.052) * | 9.841 | 1 | .002 |
| | Negative | .379(.045) * | .222(.045) * | 4.300 | 1 | .038 |
| **Narcissism** | | .070(.045) | .268(.042) * | 5.974 | 1 | .015 |
| **Locus of Control** | Internality | .074(.050) | .018(.051) | 0.494 | 1 | .482 |
| | Chance | .233(.048) * | .321(.045) * | 0.893 | 1 | .344 |
| | Powerful Others | .273(.043) * | .303(.044) * | 0.061 | 1 | .806 |
| **Cognitive Reflection Test** | Correct | .089(.051) | -.332(.052) * | 18.428 | 1 | < .001 |
| **R-squared** | | .300(.033) * | .371(.032) * | 0.146 | 1 | .703 |

Note

* p < .05.

psychological constructs are shown in S1 Table. Partial correlations between belief systems and psychological constructs for the three studies. The regression model and Wald tests of whether the predictor variables are differentially associated with the two types of belief systems are shown in Table 6.

The results for self-esteem and locus of control were broadly consistent with the previous studies, with negative self-esteem associated with paranoia (with a significantly less but nonetheless significant effect for conspiracy theories), positive self-esteem specifically associated with conspiracy theories, and the chance and powerful others locus of control variables associated with both belief systems. Importantly, as predicted, narcissism was specifically associated with conspiracy theories and, consistent with the findings from Study 2, poor analytic thinking was also specifically associated with conspiracy theories.

Contrary to prediction there was no association between either of the ECR attachment measures and either of the two belief systems.

## Study 3 discussion

The general picture that emerged from this study is consistent with the findings from the two previous studies. A two factor CFA model was the best fit to the data, despite our assiduous efforts to eliminate possible method effects. Paranoia was more associated with negative self-esteem than conspiracy theories. The observation that conspiracy theories were specifically associated with narcissism is consistent previous studies [31, 39], and helps to explain the association also observed with positive self-esteem. We discuss this finding and also the replicated finding of a specific association between conspiracy theories and poor analytical thinking in the general discussion below. The major unexpected finding from this study was the lack of association between either belief system and the two attachment scales. This is unlikely to be an artefact of our analytic approach as inspection of S1 Table in the supplementary materials all the relevant bivariate correlations were close to zero.

## General discussion

In this paper, we have reported three studies of the relationship between paranoia and conspiracy theories, one with a largely student population and two with a larger samples that were much more representative of the UK population. Many, although not all of the results were broadly consistent across the studies.

**Table 7. Summary of associations between paranoia and belief in conspiracy theories and psychological constructs in Studies 1 – 3.** Upper row for each construct shows direction of association (positive, negative or not significant) and lower row shows whether a significant difference between paranoia and conspiracies. Note that this table does not show the magnitude of the effects.

| | | Study 1 | | Study 2 | | Study 3 | |
|---|---|---|---|---|---|---|---|
| | | Paranoia | CTs | Paranoia | CTs | Paranoia | CTs |
| Insecure attachment | Anxious | + | + | + | ns | ns | ns |
| | | P > C | | P > C | | No difference | |
| | Avoidant | + | + | + | ns | ns | ns |
| | | No difference | | P > C | | No difference | |
| Self-esteem | Positive | - | ns | ns | + | - | + |
| | | C > P | | C > P | | C > P | |
| | Negative | + | ns | + | ns | + | + |
| | | P > C | | P > C | | P > C | |
| Narcissism | | | | | | ns | + |
| | | | | | | C > P | |
| Locus of control | Internality | - | ns | ns | - | ns | ns |
| | | No difference | | P>C | | No difference | |
| | Chance | + | + | + | + | + | + |
| | | P > C | | C > P | | No difference | |
| | Powerful others | + | + | + | + | + | + |
| | | No difference | | No difference | | No difference | |
| Cognitive reflection (errors) | | | | ns | + | ns | + |
| | | | | C > P | | C > P | |
| Loneliness | | + | + | + | + | | |
| | | P > C | | No difference | | | |

In all three studies we found that paranoia and conspiracist thinking, although correlated to a similar extent found in previous studies [3], were separable psychological phenomena. In our confirmatory factor analyses, models with two correlated factors were consistently far better fits to the data than single factor models. This finding was upheld despite the fact that, across the three studies, we employed two separate measures of conspiracist thinking, the Conspiracy Mentality Questionnaire [22] and the Generic Conspiracist Beliefs Scale [65], and two measures of paranoia (a version of the Persecution and Deservedness Scale [49] and a revised version of the scale designed especially for this research) and despite the fact that we made assiduous attempts to eliminate potential method effects in the final study.

The findings on the relationships between the two types of belief and our psychological constructs are summarised in Table 7, and, as would be expected for distinct but correlated phenomena, these point to both common and specific factors. The most striking common factor is that both are associated with an external locus of control and, specifically, the belief that life is dominated by chance factors and the actions of powerful others. These findings are broadly consistent with the results from previous studies of paranoia [14] and conspiracy theories [28] and can be understood in the context of research that has shown that experiences of low control lead to belief system justification and more extreme attitudes [71, 72].

Both belief systems were also associated with loneliness although, in one study (Study 1, with the least representative sample), this effect was greater for paranoia. Despite some inconsistencies, the remaining findings support our hypothesis, and that of previous researchers [3], that paranoia is specifically associated with interpersonal vulnerability. This is particularly evident in the self-esteem data, for which, across all three studies, paranoia is associated with high negative self-esteem and low positive self-esteem and, less consistently, in the attachment data.

Our findings for paranoia and negative self-esteem are consistent with a large number of previous studies that have employed nonclinical and clinical samples [15]. By contrast, in the two studies reported here that included the most representative samples, we observed positive associations between positive self-esteem and belief in conspiracy theories and, in the final study, also a positive association between conspiracy theories and narcissism. Reviewing previous studies of conspiracy theories and self-esteem, Cichocka et al. [30] noted that findings had been inconsistent, leading them to propose that conspiracy theories are associated with an excessive sense of self-worth linked to narcissism; in three studies they observed the predicted association between narcissism and conspiracy theories, which we replicated in our Study 3, but they also found that self-esteem alone was not associated with conspiracist thinking. An important difference in our studies is that we chose scales that distinguished between positive and negative self-esteem whereas previous studies, including those by Cichocka et al., used unidimensional scales. Our choice partly reflected our clinical background in which the distinction is meaningful in the context of psychopathological states; although historically it has been contested [73] the distinction has been supported in factor analytic investigations of both the Core Self-Schema Scale used in Study 1 [51] and the Self-Esteem Rating Scale [66] used in Studies 2 and 3. Overall, our findings are therefore consistent with Cichocka et al's hypothesis that conspiracy theories are associated with an inflated sense of self-worth.

Previous research has repeatedly reported robust associations between insecure (especially anxious) attachment and paranoia in both healthy and clinical (psychosis) samples [16, 17]; a recent systematic review found that 11/12 clinical studies observed this effect [74]. Two studies have reported a similar association with conspiracy theories [20, 33]. In two of the present studies, including Study 2 which had the largest and most representative sample, anxious attachment was more highly associated with paranoia than conspiracy theories and, in one, the avoidant style was also more highly associated with paranoia. This picture is complicated by the results of Study 3, which used a short version of the Experience of Close Relations Scale rather than the Relationship Scale employed in Studies 1 and 2, and in which no association was observed between either anxious or avoidant attachment and either paranoia or conspiracy theories. It is unlikely that the change of scale can explain these null results, as the two previous studies that have reported associations between attachment and conspiracy theories used an earlier variant of the RQ [20] and a variant of the ECR [33]. We therefore frankly state that we have no explanation for the null results for attachment in Study 3. However, on balance, both in the existing literature and the studies reported here, insecure attachment, especially of the anxious variety, seems more clearly associated with paranoia than with conspiracist thinking.

Finally, we included versions of the Cognitive Reflection Test in two of our studies and, against expectation, in both found evidence that poor performance was associated with conspiracy theories but not paranoia. This observation is striking given that clinical paranoia is associated with poor performance on cognitive tests related to executive function [18], and that poor analytic reasoning strongly predicts the 'jumping-to-conclusions' data gathering bias [75] which has been widely replicated in deluded patients [19]. A possible explanation is that paranoid beliefs are typically idiosyncratic whereas other kinds of strong beliefs, including conspiracy theories, are typically shared and transmitted socially [76]. It has been argued that one feature of conspiracy theories is that they can usually be rejected by reflection and simple thought experiment [77]; for example, faking the Moon landing would require 46,000 NASA employees to repeat the same lie for decades, something which, arguably, would be more difficult to engineer than actually going to the Moon. One function of analytic thinking is the detection of implausible or pseudo-profound ideas [78] and fake news [79], a skill that is

possibly less likely to impact on the evaluation of self-generated beliefs than on the evaluation of theories transmitted from others.

We acknowledge some limitations of the studies reported here. Although the reliability coefficient of most of the scales in this study were good, the locus of control subscales had only moderate reliability. The sample in Study 1 was not representative of the general population. Although Studies 2 and 3 were much more representative of the British population (and larger than any previous studies addressing the relationship between the two belief systems), and although the correlations we observed in our data were similar to those observed in studies in North America and continental Europe, generalizability to other countries and cultures should not be assumed. Indeed, it seems very likely that both belief systems will vary with personal, economic and historical circumstances. In the case of paranoia, there is considerable evidence that early life adversity plays an important role [11]. The observation that belief in conspiracy theories can be reinforced by loss of control experiences [28]. suggests a mechanism by which economic threats can lead to conspiracist thinking. Finally, it is important to note that all three studies were cross sectional.

A heuristic model that synthesizes the findings from the studies is shown in Fig 1. It should be noted that this model is structural, identifying common and specific psychological features of paranoia and conspiracy theories, and we make no strong claims about the causal role of the psychological processes identified. However, some findings relevant to this issue have been reported in previous studies. For example, it has been shown that changes in negative self-esteem predict changes in paranoia in the short [80] and long-term [81] and that priming uncertainty exacerbates belief in conspiracy theories in the short term [28]. Further experimental and longitudinal studies are required to explore the role of common and specific psychological mechanisms in causing paranoia and conspiracy theories.

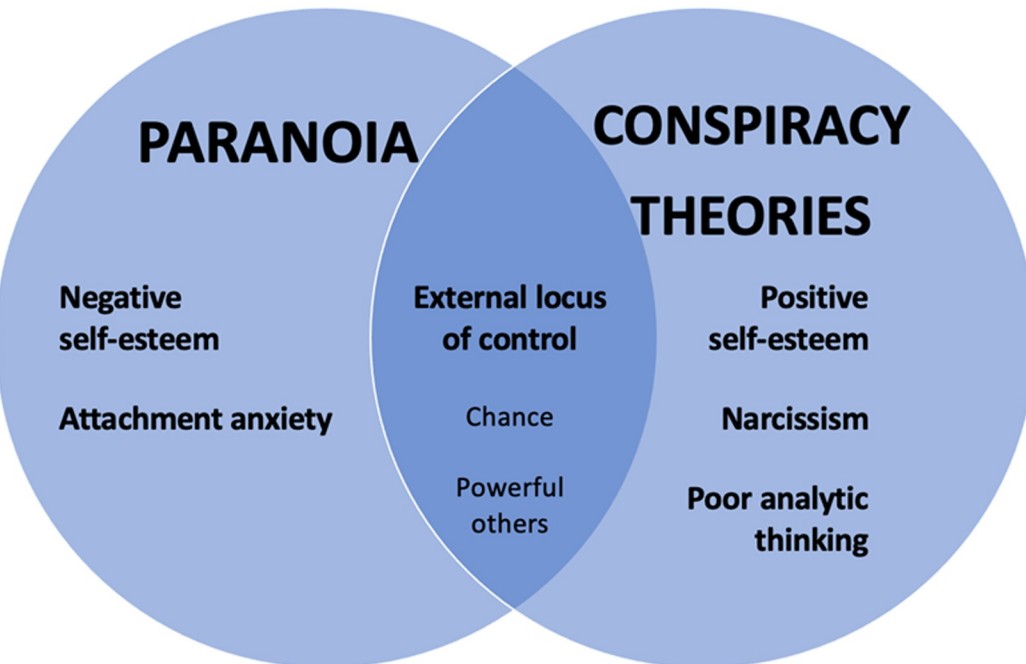

**Fig 1. Conceptual diagram showing relationship between paranoia and conspiracy mentality in relation to relevant psychological constructs**

In conclusion, in three samples (two representative of the UK population), we have shown that paranoia and conspiracy theories are separate albeit correlated psychological phenomena and characterised their psychological covariates using exacting statistical methods that have allowed us to understand which type of belief each covariate is most closely related to. In principle, the correlation between the two belief systems might reflect either shared causal determinants (for example, the perception of being powerless and having little control) leading to both or one type of belief leading to the other (for example, when someone's paranoia leads them to develop conspiracy theories) and further research is required to explore each of these possibilities. It would be useful, for example, to explore longitudinal changes in paranoia and conspiracy theories over time, identify circumstances and situations that provoke one or the other, and also compare those who show evidence of both kinds of belief systems with those who report only one kind of belief system.

Our findings have a number of important implications and point to the importance of further disambiguating the relationship between the two belief systems. First, confusion between the two is common in both the academic world and in everyday life; indeed Hofstadter's [1] misattribution of conspiracy theories to paranoia has been repeated by more recent historians [82]. Second, the failure to distinguish between the two may compromise research into both of the belief systems, leading to psychological and social predictors being attributed to the wrong belief system; indeed, it is advisable for researchers to include measures of both types of belief in their studies wherever possible. Third, only one of these belief systems (paranoia) is normally regarded as a target for therapeutic intervention whereas the other (conspiracy theories) is normally considered to be a matter of social concern; a better understanding of the relationship between the two will assist in deciding what kind of intervention is appropriate in what circumstances.

## Supporting information

**S1 Table. Partial correlations between belief systems and psychological constructs for the three studies.**
(DOCX)

**S1 Dataset.**
(SAV)

**S2 Dataset.**
(SAV)

**S3 Dataset.**
(SAV)

## Author Contributions

**Conceptualization:** Azzam Alsuhibani, Daniel Freeman, Richard P. Bentall.

**Formal analysis:** Azzam Alsuhibani, Mark Shevlin, Richard P. Bentall.

**Investigation:** Azzam Alsuhibani, Daniel Freeman, Bryony Sheaves.

**Methodology:** Mark Shevlin.

**Project administration:** Bryony Sheaves.

**Writing – original draft:** Azzam Alsuhibani, Richard P. Bentall.

**Writing – review & editing:** Mark Shevlin, Daniel Freeman, Bryony Sheaves, Richard P. Bentall.

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
