## [Decision Letter · Decision Letter 0]

18 Jun 2021

PONE-D-21-14707

Why Conspiracy Theorists Are Not Paranoid: Conspiracy Theories and Paranoia Form Separate Factors with Distinct Psychological Predictors

PLOS ONE

Dear Dr. Bentall,

Thank you for submitting your manuscript to PLOS ONE. After careful consideration, we feel that it has merit but does not fully meet PLOS ONE’s publication criteria as it currently stands. Therefore, we invite you to submit a revised version of the manuscript that addresses the points raised during the review process.

We look forward to receiving your revised manuscript.

Kind regards,

Peter Karl Jonason

Academic Editor

PLOS ONE

Journal Requirements:

3. Please change "female” or "male" to "woman” or "man" as appropriate, when used as a noun (see for instance https://apastyle.apa.org/style-grammar-guidelines/bias-free-language/gender).

Reviewers' comments:

Reviewer's Responses to Questions

**Comments to the Author**

1. Is the manuscript technically sound, and do the data support the conclusions?

Reviewer #1: Yes

Reviewer #2: Yes

2. Has the statistical analysis been performed appropriately and rigorously? 

Reviewer #1: Yes

Reviewer #2: Yes

3. Have the authors made all data underlying the findings in their manuscript fully available?

Reviewer #1: No

Reviewer #2: Yes

4. Is the manuscript presented in an intelligible fashion and written in standard English?

Reviewer #1: Yes

Reviewer #2: Yes

5. Review Comments to the Author

Reviewer #1: This is a very interesting paper that provide an insightful comparison between subclinical paranoia and conspiracy beliefs. I would reccomend it for acceptance.

I have just a minor observation:

1. In Study 2, you reported that the GCB has 17 items, but in the Study 3 it has 15 items. I think the latter is the correct one.

Reviewer #2: Thank you for the opportunity to review this manuscript, in which the authors further investigate the distinction between paranoia and belief in conspiracy theories. In three cross-sectional studies, they found a two-factor model of paranoid beliefs and conspiracy belief to be a better bit than a one-factor model, and that both belief systems have distinct and similar psychological factors associated with them.

Overall, I think that the manuscript reads well, the findings were very interesting, and it provides a meaningful contribute to our understanding of the relationship between paranoid beliefs and conspiracy beliefs. I have only some minor points that can be addressed:

Introduction:

There have been theoretical advancements in the conspiracy literature that were not included in the introduction. This was surprising considering the predictors variables that were included in the studies. People may be motived to believe in conspiracy theories (often unconsciously) in an attempt to satisfy/defend important psychological needs: existential (e.g., attachment insecurities), epistemic (e.g., intuitive vs analytical), social (e.g., self-esteem, narcissism; Douglas et al., 2017; see also van Prooijen, 2020). I would suggest including this framework in the conspiracy section of the introduction.

Discussion:

Paranoia and belief in conspiracy theories are two distinct psychological phenomena, agreed. What does this mean for the relationship between them though? Your title states that “..conspiracy theorists are not paranoid”. Does this mean that if someone believes in conspiracy theories for paranoid reasons (i.e., that everyone is out to get them), then they are not a conspiracy theorist but just paranoid? I think a lot more can be said in the discussion to tease apart the nuance between belief in conspiracy theories and paranoia.

The discussion lacks a conclusion to tie it all in. In light Imhoff and Lamberty (2018) and the current findings, what does this mean for the relationship between paranoia and conspiracy belief going forward?

6. PLOS authors have the option to publish the peer review history of their article (what does this mean?). If published, this will include your full peer review and any attached files.

Reviewer #1: No

Reviewer #2: No

---

## [Author Response · Author response to Decision Letter 0]

8 Oct 2021

Reviewer #1: This is a very interesting paper that provide an insightful comparison between subclinical paranoia and conspiracy beliefs. I would reccomend it for acceptance.

I have just a minor observation:

1. In Study 2, you reported that the GCB has 17 items, but in the Study 3 it has 15 items. I think the latter is the correct one.

RESPONSE: This discrepancy was explained in footnotes in our originally submitted manuscript: the two extra items in study 2 originate in a study of Wood et al. (2012). We have moved this explanation into the description of the scale. A footnote in the results section, explaining that we have indeed replicated Wood et al., is now included in the main text of the results section of Study 1.

Reviewer #2: Thank you for the opportunity to review this manuscript, in which the authors further investigate the distinction between paranoia and belief in conspiracy theories. In three cross-sectional studies, they found a two-factor model of paranoid beliefs and conspiracy belief to be a better bit than a one-factor model, and that both belief systems have distinct and similar psychological factors associated with them.

Overall, I think that the manuscript reads well, the findings were very interesting, and it provides a meaningful contribute to our understanding of the relationship between paranoid beliefs and conspiracy beliefs. I have only some minor points that can be addressed:

Introduction:

There have been theoretical advancements in the conspiracy literature that were not included in the introduction. This was surprising considering the predictors variables that were included in the studies. People may be motived to believe in conspiracy theories (often unconsciously) in an attempt to satisfy/defend important psychological needs: existential (e.g., attachment insecurities), epistemic (e.g., intuitive vs analytical), social (e.g., self-esteem, narcissism; Douglas et al., 2017; see also van Prooijen, 2020). I would suggest including this framework in the conspiracy section of the introduction.

RESPONSE: We have rewritten the introductory summary of the literature on conspiracy theories, organizing it around the three factor account of Douglas et al. (2017).

Discussion:

Paranoia and belief in conspiracy theories are two distinct psychological phenomena, agreed. What does this mean for the relationship between them though? Your title states that “..conspiracy theorists are not paranoid”. Does this mean that if someone believes in conspiracy theories for paranoid reasons (i.e., that everyone is out to get them), then they are not a conspiracy theorist but just paranoid? I think a lot more can be said in the discussion to tease apart the nuance between belief in conspiracy theories and paranoia.

The discussion lacks a conclusion to tie it all in. In light Imhoff and Lamberty (2018) and the current findings, what does this mean for the relationship between paranoia and conspiracy belief going forward?

RESPONSE: We slightly have slightly changed the title so it now reads ‘Why conspiracy theorists are not always paranoid’. We have also added two substantial paragraphs to the conclusion in which we summarise our findings, address the relationship between paranoia and conspiracy theories (including the possibility that paranoia might lead to CTs) and highlight three important theoretical, practical and research implications for studies of the two belief systems going forward.

---

## [Editor Report · Decision Letter 1]

12 Oct 2021

Why Conspiracy Theorists Are Not Always Paranoid: Conspiracy Theories and Paranoia Form Separate Factors with Distinct Psychological Predictors

PONE-D-21-14707R1

Dear Dr. Bentall,

We’re pleased to inform you that your manuscript has been judged scientifically suitable for publication and will be formally accepted for publication once it meets all outstanding technical requirements.

Kind regards,

Peter Karl Jonason

Academic Editor

PLOS ONE
---

## [Editor Report · Acceptance letter]

20 Dec 2021

PONE-D-21-14707R1 

Why conspiracy theorists are not always paranoid: Conspiracy theories and paranoia form separate factors with distinct psychological predictors 

Dear Dr. Bentall:

I'm pleased to inform you that your manuscript has been deemed suitable for publication in PLOS ONE. Congratulations! Your manuscript is now with our production department. 

Kind regards, 

on behalf of

Dr. Peter Karl Jonason 

Academic Editor

PLOS ONE